# Air Pollution and Health in Africa: The Burden Falls on Children

**Courage Mlambo** [1,*] , **Phillip Ngonisa** [2], **Bhekabantu Ntshangase** [1], **Nomusa Ndlovu** [2] **and Bongekile Mvuyana** [1]

1   Faculty of Management Sciences, Mangosuthu University, Durban 4301, South Africa
2   Department of Economics, University of Fort Hare, East London 5700, South Africa
*   Correspondence: mlamboct@gmail.com

**Abstract:** This study sought to examine the impact of air pollution on health in Africa. Air pollution is a major public health concern around the world. Exposure to air pollution has been linked to a slew of negative health consequences, ranging from subclinical effects, physiological changes in pulmonary functions and the cardiovascular system, to clinical symptoms, outpatient and emergency-room visits, hospital admissions, and finally to premature death. Health impact assessments indicate that sub-Saharan Africa suffers a high burden of disease and premature deaths, attributable to environmental pollution in the world. The health and economic repercussions of rapid pollution increases could jeopardize African efforts to enhance economic development, establish human capital, and achieve the Sustainable Development Goals (SDGs). Despite all these, literature on pollution and health is still scanty in the case of Africa. This study was quantitative in nature, and it used a difference GMM approach to estimate its model. The GMM results showed that children are negatively affected by pollution. Children under the age of five are especially at risk, and the effects are believed to be most severe in developing countries, where exposure to high levels of ambient air pollution throughout childhood is thought to lower total life expectancy by an average of 4–5 years. Based on the findings of this study, it is recommended that African countries must not overlook the pollution problem. They must promote and use low carbon technologies and services. In the absence of active intervention, pollution will soon raise morbidity and death.

**Keywords:** air pollution; health; carbon emissions; particulate matter (PM); nitrogen dioxide





## 1. Introduction

Air pollution is a major public health concern around the world (Chen et al. 2018a; GAHP 2019; WHO 2022; European Environmental Agency 2023). Exposure to air pollution exacerbates and contributes to a variety of negative health effects. Particulate matter (PM) and nitrogen dioxide are the most widely discussed pollutants in the context of air quality management, but because air pollution is a complex mixture of gases, other pollutants may also have an impact on health (Centre for Research on Energy and Clean Air 2023; European Environment Agency 2023). Exposure to air pollution has been linked to a slew of negative health consequences, ranging from subclinical effects, physiological changes in pulmonary functions and the cardiovascular system, to clinical symptoms, outpatient and emergency-room visits, hospital admissions, and finally to premature death (Lee et al. 2014; Boogaard et al. 2019; United States Environmental Protection Agency 2022; Imperial College London 2023). It is also acknowledged that the consequences for human health are significant and extensive. Pollution is the leading environmental cause of disease and premature death. Air pollution is estimated to have caused 6.5 million fatalities worldwide in 2015, accounting for 16 percent of all deaths (European Commission 2018; World Bank 2022).

Due to the deleterious effects of air pollution, regulations have been enacted at the international, national, regional, and local levels to reduce pollution. For instance, the United Nations General Assembly endorsed the 2030 Agenda for Sustainable Development

in September 2015. The Agenda makes several references to air pollution under Targets 3.9 (to substantially reduce the number of deaths and illnesses from hazardous chemicals and air, water and soil pollution, and contamination) (WHO 2022). In 2019, the UN called on each city, region, and country to commit to "achieving air quality that is safe for its citizens, and to align its climate change and air pollution policies, by 2030"—and to do so in the name of its citizens' health (Pan American Health Organisation 2019). This demonstrates that air pollution has acquired relevance and attention on global agendas. Furthermore, widespread recognition of pollution and climate change as central public health concerns is essential for delivering a quick response and an institutional capacity that enables governments to incorporate and adopt air pollution and climate change policies in their development strategies.

Despite the global recognition that pollution has received, almost the entire global population (99%) breathes air that exceeds WHO air quality limits and threatens their health (WHO 2022). This, then, indicates that pollution is an enormous and under-addressed public health issue (GAHP 2019). This is particularly true in developing countries. People in developing countries are the most vulnerable to air pollution, but little is being done to combat it. As a result of overcrowding and unregulated urbanization, as well as the rise of industrialisation, the air pollution problem is particularly significant in developing countries. Juginović et al. (2021) argue that, while air pollution is a major public health concern in industrialized countries, it is even more pronounced in developing countries, where rapid population growth combined with widespread industry has resulted in cities with poor air quality, posing a substantial health risk. This health and social crisis is compromising people's ability to take control over their health and lives (WHO 2022; Zhao and Su 2023).

While air pollution is a global health issue, low- and middle-income countries, particularly in Sub-Saharan Africa, are more likely to be affected because of the high exposure of the population (GAHP 2019; Cai et al. 2021). Ambient air pollution is increasing across Africa and it is now the second largest cause of death in Africa (Fisher et al. 2021). Health impact assessments (Coker and Kizito 2018; Fisher et al. 2021) indicate that Sub-Saharan Africa suffers from a high burden of disease and premature death, attributable to environmental pollution in the world. The health and economic repercussions of rapid pollution increases could jeopardize African efforts to enhance economic development, establish human capital, and achieve the Sustainable Development Goals (SDGs). In the absence of active intervention, pollution will raise morbidity and death, lower economic productivity, and stifle development. In order for African countries to make strategy interventions, reliable data and reliable scholarly information is needed to create and adopt policies that can reduce air pollution and deliver national development priorities and climate goals.

Despite these hazardous effects of air pollution, studies that have looked at pollution and health in the case of Africa are still scarce. Of the few studies that were done in Africa (SSA), one looked at trade and pollution (Adams and Opoku 2020) while Katoto et al. (2019) carried out a systematic literature review. Thus, this study seeks to increase local knowledge on pollution and health by examining the impact of air pollution on health in Africa. The study shall also contribute to both policy and literature by quantifying the health impacts of air pollution. This is essential for comprehending health burdens and assessing policy options in African nations. This study makes the case that a greater comprehension of the health effects of air pollution can lead to improved policies that more precisely and effectively target particular sources of air pollution.

## 2. Literature Review

The toxicological impact of air pollution on human health is well documented in the literature, albeit mostly in developed countries. The literature identifies various mechanisms through which air pollution affects human health. Studies show that ambient pollution affects various physiological human functions, including reproduction (Checa et al. 2016; Talaulikar 2020), the respiratory system (Tie et al. 2009; Gouveia et al. 2018; Viegi et al. 2020), the cardiovascular system (Mannucci et al. 2019; Miller and Newby 2020), throm-

bosis (Nemmar et al. 2002; Baccarelli et al. 2008; Franchini et al. 2012), mental imbalances (Buoli et al. 2018; Chen et al. 2018a; Kanner et al. 2021; Qiu et al. 2022), and morbidity and mortality (Hales et al. 2021; Khojasteh et al. 2021; Khomenko et al. 2021) across different phases of human life.

In relation to the prenatal phase, the literature shows the effects of pollution on foetal growth. For instance, Leung et al. (2022) show compelling evidence of the impact of air pollution on the growth of foetuses. The authors show that an increase in pollution levels is associated with smaller ultrasound parameters. Their findings imply that an increase in PM2.5 leads to a low $z$-score among anatomic scans. The findings of Leung et al. (2022) confirm the findings of Bekkar et al. (2020), which suggest that there is a positive relationship between air pollution and impaired foetal growth. Bekkar et al. (2020) further show that pollution affects foetal growth weight, which is a key developmental indicator for perinatal morbidity and mortality, and eventually affects later life and cardiometabolic outcomes. Clemente et al. (2017) show that exposure of pregnant individuals to pollution is linked with changes in the speed of foetal formation and development, and maternal physiology. This is supported by Korten et al. (2017) and Klepac et al. (2018) who also suggest that air pollution leads to foetal growth complications. Klepac et al. (2018) further show that high exposure to air pollution during pregnancy is linked to increased respiratory complications and asthma, as well as decreased lung function in infancy and childhood.

In addition, Singh et al. (2019) investigated the impact of ambient pollution on child health and found that an increase in pollution adversely affects the child's height-for-age and weight-for-age, and the effect is more acute in poor families. Kurata et al. (2020) find that pollution heterogeneously affects the exposed individual depending on gender. In this regard, Kurata et al. (2020) find that pollution (measured by solid fuels) positively influences respiratory illness in girls but not in boys. Kurata et al. (2020) show that an increase in exposure to PM2.5 during the prenatal phase correlates with stunted growth in boys, but pollution negatively affects both boys and girls in the postnatal phase. The World Health Organization (2018) shows that air pollution has adverse effects on children. Gouveia et al. (2018) show that pollution leads to respiratory-related deaths in both infants and children. These findings confirm the findings of Mentz et al. (2018), which show that an increase in pollutants leads to increases in the shortness of breath, chest tightness, and coughs for children highly exposed to pollution. Soh et al. (2018) suggest that prenatal exposure to pollution has an enhanced effect on the childhood respiratory system. This is in agreement with Karimi and Shokrinezhad (2020), who articulate that an increase in pollution levels is associated with infant mortality. Goyal and Canning (2017) assert that high exposure to pollution in utero is linked, in children, to stunted growth and being underweight.

The detrimental effects of air pollution are also evidenced in damage to the foetus' liver and jaundice. According to Pejhan et al. (2019), higher levels of pollution (PM1, PM2.5, and PM10) are associated with increased levels of alanine aminotransferase, alkaline phosphatase, and gamma-glutamyl transferase. Their results imply that an increase in particulate matter is associated with liver injury or disease in new-born babies. Zhang et al. (2019) show that there is a positive relationship between poor quality air and neonatal jaundice.

Tie et al. (2009) investigated the long-term series of aerosol optical extinction coefficients and found statistical evidence for a relationship between air pollution and lung cancer, which eventually leads to mortality. In the same vein, Forastiere (2004) shows that air pollution emits fine particles, which easily penetrate in human lungs, resulting in lung cancer tumours. Guo et al. (2020) also show the negative effects of pollution on health. In particular, Guo et al. (2020) found a significant positive relationship between particulate matter (PM2.5) and incidences of lung cancer for both males and females in China. This is validated by the findings of Guo et al. (2016) and Han et al. (2017), who also show that air pollution leads to lung cancer. Lipfert and Wyzga (2020) show that there is a positive association between ambient air quality and lung cancer.

Studies also show that there is an association between fertility and air pollution, which acts as an endocrine disruptor, increasing oxidation stress and exerting genotoxic effect. For instance, Zhou et al. (2014) assert that an increase in particulate matter reduces the quality of semen and leads to infertility. Carré et al. (2017), Broekmans and Fauser (2010), and Talaulikar (2020) show evidence that exposure to air pollution negatively affects the reproductive capacity and leads to infertility. Alaee (2018) asserta that air pollutants effectively lead to human infertility and, if not properly controlled, can significantly influence infertility rates in the future. This is supported by Mahalingaiah et al. (2016) and Conforti et al. (2018), who give further evidence of the adverse effects of pollution on fertility in both males and females.

In adults, studies show that air pollution is linked to cardiovascular dysfunctions, respiratory disorders, neuropsychiatric complications, and dermatological effects (Tie et al. 2009; Ngoc et al. 2017; Wu et al. 2019; Miao et al. 2021). Rajagopalan et al. (2018) assert that a percentage increase in PM2.5 leads to an increase in acute cardiovascular diseases and mortality by three percent within few days. Their findings find support in the literature. For instance, Miller and Newby (2020) show that particulate matter substantially leads to cardiovascular morbidity and higher mortality rates. Tibuakuu et al. (2018) show that the effects of pollution on cardiovascular dysfunctions are stronger for susceptible groups such as the elderly, smokers, and women, among others. This corroborates with the findings by Hadley et al. (2018), Vidale and Campana (2018), and Wu et al. (2019), which show that an elevated concentration of particular matter is associated with cardiovascular diseases and mortality. Furthermore, Ghorani-Azam et al. (2016) articulate that long-term exposure to PM2.5 concentration is associated with increased risks of cardiovascular dysfunction, particularly in adults aged between 35–70 years.

In addition, air pollution is related to respiratory complications. Sciaraffa et al. (2017) show that there is a positive association between atmospheric pollutants (PM10, PM2.5, $NO_2$, $SO_2$, $CO_2$, $O_3$) and respiratory diseases. Borghini et al. (2017) further show that ambient pollution leads to a wet cough, asthma, and reduction in volumes and respiratory flows. These authors further show that air pollution increases hospital admission days. In the same vein, Lawin et al. (2016) found that high carbon dioxide exposures lead to high phlegm levels. This is supported by Arulprakasajothi et al. (2018), who show that ambient air pollutants cause respiratory disorders such as asthma, bronchitis, and emphysema. Zhu et al. (2019) argue that air pollution is linked to lung cancer and respiratory diseases, and females are more susceptible to respiratory mortality compared to their male counterparts. Similarly, Kim et al. (2018) show that ambient pollution leads to lung infection. In a recent study, Niu et al. (2021) show that exposure to ambient pollution positively leads to asthma and chronic pulmonary obstruction. Kim et al. (2018) further suggest that air pollution has acute effects on outpatients and hospital admissions. This is confirmed by Seposo et al. (2021), who state that an increase in ambient pollution leads to respiratory complications.

The literature further shows that air pollution leads to blood clots inside blood vessels. Robertson and Miller (2018) show that pollution positively leads to thrombosis. The authors further assert that acute exposure to particulate matter leads to a shift in the haemostatic balance towards a pro-thrombotic/procoagulative state and eventually leads to cardiovascular dysfunction. This is similar to Renzi et al. (2022), Hamanaka and Mutlu (2018), Zhang et al. (2019), and Miao et al. (2021), who find that pollution leads to thrombosis and inflammation, which leads to cardiovascular dysfunction. Chen et al. (2004) posit that air pollution leads to morbidity, mortality, and hospital admissions.

Evidence further shows that air pollutants are negatively related to the mental wellness of individuals, resulting in psychological disorders. Gu et al. (2020) argue that air pollution is linked with a high prominence of negative emotions such as depression, powerlessness, nervousness, and restlessness. The authors further assert that pollution increases psychological disorders, mainly in women. These findings are similar to the findings of Chen et al. (2018b), Ghorani-Azam et al. (2016), Bernardini et al. (2020), and Gu et al.

(2020). Furthermore, the literature shows that air pollution leads to depression and suicide (Buoli et al. 2018; Gładka et al. 2018; Chirico and Magnavita 2019). Buoli et al. (2018) show that air pollution potentially worsens mental conditions. Chen et al. (2018b) found a positive and significant relationship between pollution and autism. This is supported by other studies (Ritz et al. 2018; Rahman et al. 2022a; Raz and Oulhote 2022; Yang et al. 2022).

## 3. Materials and Methods

### 3.1. Data Sources

The study used a panel data set covering 50 African countries, over the period of 1970–2019. Data were sourced from the World Bank's World Development indicator database.

### 3.2. Estimation Techniques

The theoretical specification of this study follows Grossman (1972) and the empirical specification follows that of Fayissa and Gutema (2005). Grossman developed the following theoretical health production function:

$$H = f(X), \tag{1}$$

where $H$, a measure of individual health output, is a function of X (a vector of individual inputs to the health production function such as income, the environment, education, etc). However, Fayissa and Gutema (2005) argue that Grossman's model was designed for analysis of health production at the micro level and might not fit well in macro level analyses. In order to switch from micro to macro analysis without losing the theoretical ground, Fayissa and Gutema (2005) modified the Grossman (1972) model and came up with the following model:

$$H = f(Y, S, V), \tag{2}$$

where $Y$ is a vector of per capita economic variables, $S$ is a vector of per capita social variables, and $V$ is a vector of per capita environmental factors. The choice of variables for estimation in this study is in line with the model developed by Fayissa and Gutema (2005). This present study estimated the following dynamic model:

$$H_{i,t} = \alpha\, H_{i,t-1} + \beta\, x_{it} + \varepsilon_{it}, \tag{3}$$

where $H_{i,t}$, is life expectancy (or infant mortality as estimated in Model 2) of country $i$ at time $t$, $H_{i,t-1}$ is its one-period lag, $x_{it}$ a vector of the explanatory variables, and $\varepsilon_{it}$ is the error term for country $i$ at time $t$. The error term is given by:

$$\varepsilon_{it} = \mu_i + v_{it}, \tag{4}$$

where $\mu_i \sim iid\,(0, \sigma_u^2)$ and $v_{it} \sim iid\,(0, \sigma_v^2)$ are independent amongst themselves and each other. In order to eliminate country-specific effects, we first take the differences of (2):

$$H_{i,t} - H_{i,t-1} = \alpha\,(H_{i,t-1} - H_{i,t-2}) + \beta\,(x_{it} - x_{it-1}) + \varepsilon_{it} - \varepsilon_{it-1}. \tag{5}$$

However, the construction of the new error term, $\varepsilon_{it} - \varepsilon_{it-1}$ is correlated with the lagged dependent variable ($H_{i,t} - H_{i,t-1}$). In order to address the endogeneity problem, Levine et al. (2000) suggest the use of instruments. This study does this by applying a GMM estimator, which operates under the assumption that the error term, $\varepsilon_{it}$, is not serially correlated and that the explanatory variables, X, are weakly exogenous. Mućk (2020) maintains that the consistency of the GMM estimator is based on the assumption that the transformed error term is not serially correlated, i.e.,:

$$\mathbb{E}(\Delta\varepsilon_{i,t},\,(\Delta\varepsilon_{i,t-2}) = 0. \tag{6}$$

This is the difference estimator. Difference GMM is so-called because estimation proceeds after first-differencing the data in order to eliminate the fixed effects. The difference GMM approach deals with this inherent endogeneity by transforming the data to remove the fixed effects. The standard approach applies the first difference (FD) transformation, which removes the fixed effect (Baum 2002).

The GMM estimator is an instrumental variable (IV) estimator, where endogenous variables are instrumented using their lagged values (internal instruments) and/or other variables (external instruments) that are only correlated with health through the endogenous variables. Following Fayissa and Gutema (2005), the Fundamental Cause Theory, and Grossman (1972), this study employs the following specifications of the Dynamic Panel Data models:

Model 1

$$LE_{i,t} = \beta_0 + \beta_1 LE_{i,t} + \beta_2 GDP_{i,t} + \beta_2 HE_{i,t} + \beta_3 EDU_{i,t} + \beta_4 FA_{i,t} + \beta_5 PM_{i,t} + \beta_6 HIV_{i,t} + \beta_7 REM_{i,t} + \mu_{i,t}, \quad (7)$$

Model 2

$$IM_{i,t} = \beta_0 + \beta_1 LE_{i,t} + \beta_2 GDP_{i,t} + \beta_2 HE_{i,t} + \beta_3 EDU_{i,t} + \beta_4 FA_{i,t} + \beta_5 PM_{i,t} + \beta_6 HIV_{i,t} + \beta_7 REM_{i,t} + \mu_{i,t}, \quad (8)$$

where LE is life expectancy (a measure for health), IM is infant mortality, GDP is economic growth, HE is health expenditure, EDU is education, FA is foreign aid, PM is particulate matter, a proxy for air pollution, HIV is human immunodeficiency virus, and REM is remittances. These variables are presented in Table 1 below.

**Table 1.** Description of variables.

| Variable Symbol | Variable Definition | Source of Data |
|---|---|---|
| LE | Life expectancy at birth. Life expectancy is a statistical measure of the estimate of the span of a life. | World Bank Development Indicators |
| IM | Infant mortality. Infant mortality rates measure child survival. The infant mortality rate is the probability of a child born in a specific year or period dying before reaching the age of one. | World Bank Development Indicators |
| GDP | Economic Growth. Annual percentage growth rate of GDP at market prices based on constant local currency. | World Bank Development Indicators |
| HE | Health Expenditure. Health expenditure includes all expenditures for the provision of health services, family planning activities, nutrition activities, and emergency aid designated for health. | World Bank Development Indicators |
| EDU | Education. Secondary school enrolment, percent of all eligible children. | World Bank Development Indicators |
| FA | Foreign aid and official development assistance received. | World Bank Development Indicators |
| PM | Particulate matter. Net official development assistance (ODA) consists of disbursements of loans made on concessional terms (net of repayments of principal) and grants by official agencies. | World Bank Development Indicators |
| HIV | Prevalence of HIV, total (% of population aged 15–49). Prevalence of HIV refers to the percentage of people aged 15–49 who are infected with HIV. | World Bank Development Indicators |
| REM | Personal remittances received (currency USD). Personal remittances comprise personal transfers and compensation of employees. Personal transfers consist of all current transfers in cash or in kind made or received by resident households to or from non-resident households | World Bank Development Indicators |

The study's main variables are life expectancy and particulate matter for Model 1 and infant mortality and particulate matter for Model 2. Life expectancy acts as a proxy for

health and particulate matter as a proxy for air pollution. This choice was guided by the literature. Life expectancy is seen as a good measure of health (OECD 2017; Chewe and Hangoma 2020). The second model was incorporated because some studies (Gouveia et al. 2018; World Health Organization 2018; Singh et al. (2019) have shown that children are more affected by pollution than adults. For example, the World Health Organization (2018) shows that air pollution has adverse effects in children. Gouveia et al. (2018) show that pollution leads to respiratory-related deaths in both infants and children. The selection of independent variables was based on the literature, particularly Fayissa and Gutema (2005), the Fundamental Cause Theory, and Grossman (1972). According to Fundamental Cause Theory, it is social inequality in access to flexible social resources (in particular, wealth, income, education, and racial privilege) that drives population health inequalities (Riley 2020). On the other hand, Grossman developed the following theoretical health production function, which depicted health as a function of X (a vector of individual inputs to the health production function such as income, the environment, education, etc.).

## 4. Results

### 4.1. Correlations

Table 2 contains the correlation coefficients between the dependent and independent variables.

**Table 2.** Correlation test.

|  | **PM** | **FA** | **GDP** | **HE** | **EDU** | **REM** |
|---|---|---|---|---|---|---|
| PM | 1 | 0.1767 | −0.0788 | 0.1496 | 0.4350 | 0.0264 |
| FA | 0.1767 | 1 | 0.0933 | −0.0710 | −0.1327 | 0.4391 |
| GDP | −0.0788 | 0.0933 | 1 | −0.1661 | −0.1504 | −0.5702 |
| HE | 0.1496 | −0.0710 | 0.1661 | 1 | 0.3182 | 0.3321 |
| EDU | 0.4350 | −0.1327 | −0.1504 | 0.3182 | 1 | 0.0123 |
| REM | 0.0264 | 0.4391 | −0.5702 | 0.3321 | 0.0123 | 1 |

The results show that the coefficients were quite low, indicating that multicollinearity was unlikely to be a problem in our estimations. The correlation coefficients were found to be below 0.8, as suggested by econometric studies.

### 4.2. Results and Discussion

The results from the empirical estimation are presented in Table 3 with the application of the difference GMM.

The results show that there is a very weak positive relationship between pollution and life expectancy. However, these results are insignificant. This means that pollution does not affect life expectancy in Africa. The results are surprising because pollution is seen as a health hazard. Ebenstein et al. (2017), Bennett et al. (2019), Lelieveld et al. (2020) and Fatima et al. (2021) note that one of the major worldwide health concerns, air pollution, is responsible for a large excess mortality and reduction in life expectancy. According to a World Bank analysis from 2020, exposure to pollution decreased the average global life expectancy at birth by almost one year in 2016. A study by Rahman et al. (2022b) showed that emissions have a significantly negative impact on life expectancy, suggesting that higher the carbon emissions lower the life expectancy. Poor air quality is an established predictor of poor health outcomes, and even small air quality improvements are estimated to prevent millions of premature deaths each year (Shindell et al. 2022; Burnett et al. 2018). However, the findings of this study are in line with those of OECD (2017) and Hailemariam and Pan (2019) and Mahalik et al. (2022). Hailemariam and Pan (2019) revealed that a 10 percent increase in per capita carbon dioxide emissions in the atmosphere causes an increase of 2.2% in the crude death rate and a reduction in life expectancy of about 0.6%.

Mahalik et al. (2022) showed that $CO_2$ emissions seem to improve the life expectancy for developing countries. In an OECD (2017) study it was found that air pollution was also not significantly associated with life expectancy gains, despite there being clear evidence elsewhere of the adverse effects of air pollution on health. This result reflects the long lag in time before air pollution affects a person's health, and also the relatively small decreases in air pollution over time (OECD 2017).

**Table 3.** Regression results: one-step GMM.

| Variables | (Model 1) LE | (Model 2) INFM |
|---|---|---|
| LE (−1) | 0.2499 ** | |
| | (0.0898) | |
| INFM (−1) | | 0.5346 ** |
| | | (0.2086) |
| EDU (Education) | 0.2756 ** | 0.5974 ** |
| | (0.0696) | (0.2674) |
| HE (Health Expenditure) | 0.0078 | −0.1023 *** |
| | (0.0701) | (0.0009) |
| GDP (Economic Growth) | 0.2788 *** | 0.1521 *** |
| | (0.1416) | (0.0043) |
| FA (Foreign Aid) | 0.1956 ** | 0.5528 |
| | (0.0822) | (0.2620) |
| PM (Particulate Matter) | 0.00377 | −0.0159 *** |
| | (0.00636) | (0.0029) |
| HIV | 0.0216 *** | 0.0628 *** |
| | (0.0007) | (0.0049) |
| REM (Remittances) | 0.2587 | 0.5251 |
| | (0.2327) | (0.5120) |
| Number of Countries | 50 | 50 |
| J-statistic | 2.441 | 3.219 |
| *(p-value)* | 0.691 | 0.487 |

Robust standard errors in parentheses. *** $p < 0.01$, ** $p < 0.05$.

However, it should also be noted that life expectancy, although it is seen as a good measure of health (Chewe and Hangoma 2020), fails to capture the health of individuals sufficiently. What needs to be considered is whether the additional years are spent in good or bad health. Many years might be lived in poor health. In 2017, life expectancy at birth in Sub-Saharan Africa was 63.9 years, but healthy life expectancy was only 55.2 years (Roth et al. 2018). This suggests that people are living many years in poor health. However, due to data issues, this study could not use the health life expectancy as a proxy for health. In order to address this shortcoming, the study had to use another health indicator: infant mortality. This indicator was used because the issue of pollution has been seen to be affecting vulnerable groups such as children (UNICEF 2017; UNEP 2022). Furthermore, infant mortality is an indicator of the state of the health care system and can signal the need for improved health care services (Reidpath and Allotey 2003; Chewe and Hangoma 2020). In order to capture this, Model 2 used infant mortality as a health indicator.

Results showed that pollution positively affected infant mortality. This is in line with literature. Manisalidis et al. (2020) found that children and the elderly are more vulnerable to the short- and long-term consequences of high levels of environmental pollution, which raises the mortality rate. Children under the age of five are especially at risk, and the effects are believed to be most severe in developing countries where exposure to high levels of ambient air pollution throughout childhood is thought to lower total life expectancy by an average of 4–5 years (Lelieveld et al. 2018; Jordan 2020; Fisher et al. 2021; Osipov et al. 2022). According to the World Health Organization (2018) and Cullillan (2023), children are known to be more vulnerable to the adverse health effects of air pollution due to their higher per minute ventilation, immature immune system, involvement in vigorous activities, and the

longer periods of time they spend outdoors. Several studies (Friedrich 2018; Heft-Neal et al. 2018; Jordan 2020; deSouza et al. 2022; Wang et al. 2023) have shown that air pollution contributes to infant mortality. A study by deSouza et al. (2022) showed that fine particulate matter in air has contributed to infant mortality in India. Heft-Neal et al. (2018) showed that mortality rates are substantially higher for infants exposed to higher levels of particulate matter. The study further showed that poor air quality is responsible for one in five infant deaths in Sub-Saharan Africa. Wang et al. (2023) showed that fine particulate matter in the air has contributed to infant mortality in China. A study conducted by the European Environment Agency (2023) linked pollution to low birth weight and premature birth and concluded that children are most vulnerable to the health impacts of air pollution.

The results show a positive relationship between education and life expectancy. This is consistent with the literature. There is a well-established, significant relationship between education and health. According to Picker (2007), an additional four years of education lowers five-year mortality by 1.8 percentage points. The Fundamental Cause Theory asserts that social variables like education are "fundamental" causes of disease and health because they affect access to a wide range of material and non-material resources like money, safe neighbourhoods, and healthier lifestyles, all of which protect or improve health (Zajacova and Lawrence 2018). Furthermore, the Human Capital Theory (HCT) states that education enhances a person's knowledge, skills, reasoning, efficacy, and a wide range of other abilities, all of which can be used to generate health (Mirosky and Ross 2005). People who are well educated experience better health, as reflected in the high levels of self-reported health and low levels of morbidity, mortality, and disability (OECD 2017; Raghupathi and Raghupathi 2020). While those with higher levels of education are frequently more aware of the benefits and drawbacks of particular actions, they are also more likely to absorb and act on this information. This demonstrates how good education serves as the cornerstone upon which health and wellness are built.

The effect of education is even more pronounced on infant mortality. Results show a positive relationship between education and infant mortality. This is consistent with the literature. The educational attainment of parents, particularly mothers, has been associated with lower levels of child mortality (Balaj et al. 2021). A study by a Schellekens (2021) also showed that better maternal education explains 15% of the infant mortality decline in Indonesia from 1980 to 2015. Education assists women in making wiser decisions regarding a variety of health and disease-related issues, including prenatal care, basic hygiene, nutrition, and immunization, all of which are crucial for lowering the primary causes of death in children under the age of five.

Foreign aid is seen to have a positive relationship with life expectancy. Foreign aid for health care is directly linked to an increase in life expectancy and a decrease in child mortality in developing countries (Bendavid and Bhattacharya 2014; Ritcher 2014). This refutes the claim that foreign aid has been associated with delayed development. The fungibility of aid, or whether it will be used for its intended purpose or be redirected to another sector or disappear owing to corruption, has a significant problem with its distribution (Van de Sijpe 2013). Martinez-Alvarez and Acharya (2012) are also of the view that aid fragmentation creates extra costs for recipient countries and reduces the effectiveness of foreign aid on health. However, this study found that foreign aid is effective in increasing life expectancy in African countries. Toseef et al. (2020) also found evidence that foreign aid improved life expectancy in developing countries. However, results show that the relationship between foreign aid and infant mortality is insignificant. This shows that foreign aid does not influence infant mortality.

Health expenditure is seen to have a positive but insignificant relationship with life expectancy. This is surprising because health expenditure must significantly contribute to health. For instance, Bein et al. (2017) found that an increase in healthcare spending is correlated with an increase in life expectancy and a decrease in new-born, under-five deaths, and neonatal fatalities. Jaba et al. (2014), Aísa et al. (2014), and Rahman et al. (2018) also revealed an association between life expectancy and healthcare expenditures. Our

findings are in tandem with Deshpande (2014), who observed that there is no significant correlation between healthcare spending and life expectancy in developing countries. They further argued that in developing countries it is not the quantity spent but the quality of expenditure that impacts healthcare. Furthermore, Zarulli et al. (2021) argues that, among countries with lower levels of education, decreasing unemployment and income inequality increases average life expectancy without increasing health expenditure levels. However, corruption, misuse, and displacement of aid from the health sector undermines the ability to use these resources for health improvements. Filmer et al. (1998) identified that, rather than public healthcare expenditure, the level of poverty, income inequality, female education, and other socio-economic factors were the main determining factors of child mortality.

However, results show that health expenditure reduces infant mortality. This shows that an increase in health expenditure reduces infant mortality. The results are consistent with empirical literature. Houeninvo (2022) showed that an increase in public health expenditure per capita by 1% reduces infant mortality and child mortality in the next 3 year period by 0.009 and 0.0268%, respectively. Dhrifi (2018) and Kiross et al. (2020) had, earlier, showed the contributory role of public health expenditure to infant mortality. However, Akinlo and Sulola (2019) argued that, due to corruption in African states, public health expenditure does not have an effect on health outcomes (including infant mortality).

GDP is found to have a positive relationship with life expectancy. The empirical relationship between mortality and national income was first noted by Samuel Preston, who found that there is a positive log relationship between a country's gross domestic product (GDP) per capita and life expectancy (Shkolnikov et al. 2019). There is a strong relationship between GDP and life expectancy (Lleras-Muney cited in (Guo et al. 2016)). Income influences the condition of people's lives and is a main socioeconomic determinant of health (Bayati et al. 2013). Sickles and Taubman (1997) had earlier provided evidence that life expectancy increases as a country improves its standard of living has long been recognized, as higher income is typically associated with development and makes possible, in part, the consumption of goods and services that improve health. The data strongly suggest that longevity is an economic good. Through increased economic growth and development in a nation, GDP per capita raises the life expectancy at birth and so contributes to the extension of lifespan (Miladinov 2020).

Remittances were seen to have a positive but insignificant relationship with life expectancy. Several studies have found the developmental impact of remittances. Ponce et al. (2011) find significant effects of remittances on medicine expenditures when illness occurs. Remittances increase healthcare access for the poor and low-income earners in developing countries (Awojobi 2020). Zhunio et al. (2012) found that remittances play an important role in improving primary and secondary school attainment, increasing life expectancy, and reducing infant mortality. However, the findings of this study are inconsistent with a UNICEF study. In Jamaica, a joint research study by UNICEF and the Government of Jamaica found no significant differences in the health outcomes of children between remittance-recipient households and those that are not in receipt of remittances. This was in spite of increased health expenditure in remittance-recipient households (UNICEF 2013).

*4.3. Diagnostics*

For the model to be valid it must be ensured that the model passes the J-test and the Arellano–Bond test for second-order serial correlation in the residuals. Results are presented in Table 4.

**Table 4.** Arellano–Bond test.

| Test Order | m-Statistic | rho | SE(rho) | Prob |
|:---:|:---:|:---:|:---:|:---:|
| AR (1) | −2.3312 | −143.34 | 82.129 | 0.0197 |
| AR (2) | 0.9847 | 64.311 | 73.97 | 0.3247 |

The results show that the first- and second-order autocorrelations reveal no evidence against the validity of the instruments used in this study. The *p*-value for AR(1) is less than 0.05 and for AR(2) it is insignificant. If the innovations are i.i.d., the first order statistic must be significant, and the second order statistic must be insignificant. The J-statistic turned out to be 2.441 and it had a *p*-value of 0.691. This shows that the instruments used in this study were valid.

*4.4. Robustness*

A simple robustness test consists of running the same model using the Fixed and Random Effects panel techniques. Results are shown in Table 5.

**Table 5.** Regression results: Random Effects and Fixed Effects.

| | **(FE)** | **(RE)** |
|---|---|---|
| **Variables** | **LE** | **LE** |
| EDU (Education) | 0.1372 ** | 0.3177 ** |
| | (0.0592) | (0.1365) |
| HE (Health Expenditure) | 0.5434 | 2.9148 ** |
| | (0.1375) | (1.2084) |
| GDP (Economic Growth) | 0.0052 | 0.0067 *** |
| | (0.0061) | (0.0012) |
| FA (Foreign Aid) | 0.6706 ** | 0.0265 |
| | (0.2144) | (0.0087) |
| PM (Particulate Matter) | 0.0315 *** | 0.3377 ** |
| | (0.0056) | (0.1314) |
| HIV | −1.7928 * | 0.2621 |
| | (0.8762) | (0.3015) |
| REM (Remittances) | 0.1406 | 0.0069 *** |
| | (0.4388) | (0.0010) |
| Hausman Test (chisq) | 12.567 | |
| *(p-value)* | 0.4792 | |
| Random Effect Appropriate | | |

Robust standard errors in parentheses. *** *p* < 0.01, ** *p* < 0.05, * *p* < 0.1.

The study findings reveal that most of the coefficients' sign and significance align with the GMM estimation. Both models show that there is a positive relationship between air pollution and life expectancy. This partially confirms the findings that were reported in the GMM estimation, which is this study's main output.

## 5. Conclusions and Recommendations

This study seeks to increase local knowledge on pollution and health by examining the impact of air pollution on health in Africa. The study was motivated by the fact that although air pollution is a problem for health worldwide, low- and middle-income nations—particularly those in Sub-Saharan Africa—are more likely to be impacted because of large population exposure. The direct impact of economic growth on environmental pollution can indirectly compromise public health. Exposure to air pollution has been linked to a slew of negative health consequences, ranging from subclinical effects, physiological changes in pulmonary functions and the cardiovascular system, to clinical symptoms, outpatient and emergency-room visits, hospital admissions, and finally to premature death. This study was quantitative in nature, and it used secondary panel data to achieve its objective.

Results showed that pollution and life expectancy are positively related. It can thus be assumed that the direct impact of economic growth on environmental pollution is yet to compromise public health in African countries on a macro-level. However, since the variable that was employed to measure health has been found to be deficient, the findings should be interpreted with caution. As a result, an emphasis should be placed more on "healthy life expectancy" than "life expectancy" in general by international statistical data repositories like the World Bank and UN. This should also apply to future studies in

this study area; they must make an attempt to use both life expectancy and healthy life expectancy as their dependent variables. Furthermore, micro studies (longitudinal studies) are required to scrutinise and vouch for the findings from secondary studies.

The results showed that pollution is affecting children. Results showed that pollution positively affects infant mortality. This is in line with the literature. A survey of literature conducted by the study showed that children and the elderly are more vulnerable to the short- and long-term consequences of high levels of environmental pollution, which raises the mortality rate. Children under the age of five are especially at risk, and the effects are believed to be most severe in developing countries, where exposure to high levels of ambient air pollution throughout childhood is thought to lower total life expectancy by an average of 4–5 years.

Based on these findings, this study recommends that governments engage in data collection on the causes of air pollution and its health effects, particularly on children and the general public. They should establish and maintain air quality monitoring systems and ascertain whether children are not dangerously exposed to pollution. The main goal should be to reduce children's exposure to air pollution. African governments should, therefore, adopt policies that are environmentally friendly: policies that promote and use low carbon technologies and services. In the absence of active intervention, pollution will soon raise morbidity and death. Air pollution issues need to be a priority in order to protect the health of children and support sustainable development for future generations.

**Author Contributions:** Conceptualization, C.M.; methodology, C.M.; software, C.M.; validation, C.M., P.N. and N.N.; formal analysis, C.M.; investigation, C.M. and P.N.; resources, B.N. and B.M.; data curation, C.M., P.N. and N.N.; writing—original draft preparation, C.M., P.N. and N.N.; writing—review and editing, C.M.; visualization, C.M, P.N. and N.N. All authors have read and agreed to the published version of the manuscript.

**Funding:** This research received no external funding.

**Institutional Review Board Statement:** Not applicable.

**Informed Consent Statement:** Not applicable.

**Data Availability Statement:** Not applicable.

**Conflicts of Interest:** The authors declare no conflict of interest.

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
