# Peer review of "Air Pollution and Health in Africa: The Burden Falls on Children"

_economies, doi:10.3390/economies11070196_

Round 1
Reviewer 1 Report
This paper estimates the effect of air pollution on the life expectancy in Africa. It is generally a well-written paper so I do not have critical concerns. However, some minor concerns are listed below for the author.
Why Africa? The author should discuss some thoughts about why they specifically choose African countries as the sample (given the data set have more country available). For example, it might be the case that due to the relative less developed level, people in Africa are more vulnerable to air pollution compared to people elsewhere. It is very important to throw attention to Africa, but the author should always make a balance between specific and general contribution.
In the model part, a function H() is used. Given that H=H(Y,S,V), how about using a VAR model (which I think is more common, though)?
Line 320-321 is a bit awkward. It can be included in the estimator tables.
English is generally fine, but the writing flow, especially in the result section, is a bit hard to read.
Author Response
Reviewer 1
This paper estimates the effect of air pollution on the life expectancy in Africa. It is generally a well-written paper so I do not have critical concerns. However, some minor concerns are listed below for the author.
Why Africa? - Line 67-80 explains why Africa was chosen. The study showed that low- and middle-income countries, particularly in Sub-Saharan Africa, are more likely to be affected because of high exposure of the population. Furthermore, we showed that ambient air pollution is increasing across Africa (Cross, 2021; UN, 2021) and it is now the second largest cause of death in Africa (Fisher et al. 2021). This explains why Africa was chosen as a case study.
In the model part, a function H() is used. Given that H=H(Y,S,V), how about using a VAR model (which I think is more common, though)? – The study used panel data. Panel data can suffer from endogeneity issues. VAR model may not be able to correct such issues. This needs GMM models. This is why the study chose to use a GMM model. The GMM estimator is an instrumental variable (IV) estimator where endogenous variables are instrumented using their lagged values (internal instruments) and/or other variables (external instruments).
Line 320-321 is a bit awkward. It can be included in the estimator tables. – sorry that was a mistake. We have since deleted that table.

Reviewer 2 Report
This article assesses the impact of air pollution on human health in Africa. The methods used are differential GMM methods combined with mathematical models to study the effects of selected factors on life expectancy or infant mortality. The results showed that pollution s positively correlated with life expectancy, but it can increase infant mortality.
However, there are still some problems in the article as follows.
1) The symbol subscripts of some chemical formulas in this paper are not standard, such as NO2 in Introduction.
2) In "2. Literature review", In the postnatal phase, air pollution manifests itself in the form of respiratory and cardiovascular diseases as well as lung cancer.” Please check whether the subject is air pollution or negative influence/adverse effect.
3) In "2. Literature review", "In adults, studies show that ambient pollution is linked to cardiovascular dysfunctions, respiratory disorders, neuropsychiatric complications and dermatological effects (Tie, Wu and Brasseur, 2009; Ngoc et al., 2017; Wu et al., 2019; Miao et al., 2021)." Please check the first sentence, because this paragraph mainly talks about air pollution.
4) In "2. Literature review", "Evidence further shows that air pollutants are negatively related to the mental wellness of individuals resulting in psychological disorders. " Is there missing a comma before "resulting"? Please check the sentence.
5) In "3.2 Estimation techniques", some variables are not explained clearly, such as s in the equation "E [??,?- s (??? − ???−1) ] = 0 for s ≥ 2; t = 3,…….,T". Another two are N and T in "The difference GMM is particularly suited for this study’s data as it performs well with a relatively large N and small T and thus fits well with our data that has N=51 and T=49 (Chewe and Hangom, 2020)."
6) In "3.2 Estimation techniques", in models 1 and 2, the selection of variables seems to have no basis.
7) In " 4.2 Results and discussion ", what do GDPSA, EXP01 and the following numbers mean?
8) It seems that the calculation results of this paper are not sufficiently relevant to the topic. The results are more inclined to the influence of selected factors on life expectancy, and there is a lack of analysis of pollution on children.
Quality of English Language of this article is above average, pretty good.
Author Response
Reviewer 2
However, there are still some problems in the article as follows.
1) The symbol subscripts of some chemical formulas in this paper are not standard, such as NO2 in Introduction. – thank you for pointing that out. We deleted the formulars.
2) In "2. Literature review", In the postnatal phase, air pollution manifests itself in the form of respiratory and cardiovascular diseases as well as lung cancer.” Please check whether the subject is air pollution or negative influence/adverse effect. – thank you for pointing that out. We saw that the line was rather misplaced and we decided to delete it.
3) In "2. Literature review", "In adults, studies show that ambient pollution is linked to cardiovascular dysfunctions, respiratory disorders, neuropsychiatric complications and dermatological effects (Tie, Wu and Brasseur, 2009; Ngoc et al., 2017; Wu et al., 2019; Miao et al., 2021)." Please check the first sentence, because this paragraph mainly talks about air pollution. – we reread the line and saw that truly the paragraph talks about air pollution. We had to delete the line.
4) In "2. Literature review", "Evidence further shows that air pollutants are negatively related to the mental wellness of individuals resulting in psychological disorders. " Is there missing a comma before "resulting"? Please check the sentence. – we proof read the line and so that a comma was indeed missing. We added a comma to that statement.
5) In "3.2 Estimation techniques", some variables are not explained clearly, such as s in the equation "E [??,?- s (??? − ???−1) ] = 0 for s ≥ 2; t = 3,…….,T". Another two are N and T in "The difference GMM is particularly suited for this study’s data as it performs well with a relatively large N and small T and thus fits well with our data that has N=51 and T=49 (Chewe and Hangom, 2020)." – thank you for pointing that out. We had to rework the equation. A more meaningful equation was added.
6) In "3.2 Estimation techniques", in models 1 and 2, the selection of variables seems to have no basis. – the basis for the selection of variables and the model was indicated in lines 240-241 and 282-283. On top of this, we also added a paragraph (line 303-318) that explained how the variables were selected.
7) In " 4.2 Results and discussion ", what do GDPSA, EXP01 and the following numbers mean? - sorry, this was an error. We have since deleted the table.
8) It seems that the calculation results of this paper are not sufficiently relevant to the topic. The results are more inclined to the influence of selected factors on life expectancy, and there is a lack of analysis of pollution on children. – we managed to add more studies that explained the effects of air pollution on children (line 384-394).
Reviewer 3 Report
The Author (s) of the paper "Air Pollution and health in Africa: The burden falls on children" by used panel data from 50 African countries to explore the relationship between air pollution and life expectancy and infant mortality. Though it is well established that air pollution has severe adverse health impacts, however, the findings of this study are surprising to me. My observation recommendations are as follows;
- Author (s) should improve the contribution section. They need to explain what is the specific contribution of this study to the overall hypothesis.
-if the wish they can arrange literature review thematically
-line 292- PM is not defined, AP is not given in any equation.
Table 1- add description of each variable used in the model
line 312- cite studies to support this criteria
- I am surprised to know that the Level of Particulate Matter (PM2.5) is not significantly related to life expectancy, however, the author (s) has provided a possible reason for this by mentioning good health expectancy. This must be discussed in the limitation of the study.
-The results of the first model reveal that mainly it is HIV that is having an impact on LE.
-How infant mortality was operationalised should be discussed, because it is creating confusion while looking at the results of the impact of health expenditures on Life expectancy and infant mortality, it should have the opposite effect on both. The results show a positive effect on both.
-line 338- how results are in line with the studies cited here, almost all studies cited do not mention child death as a consequence of PM emission. while this study shows a positive association between infant mortality and PM level.
-author (s) must cite studies supporting this result.
-limitation, future directions, and implication section must be added in revised version.
Moderate English revision is required.
Author Response
Reviewer 3
The Author (s) of the paper "Air Pollution and health in Africa: The burden falls on children" by used panel data from 50 African countries to explore the relationship between air pollution and life expectancy and infant mortality. Though it is well established that air pollution has severe adverse health impacts, however, the findings of this study are surprising to me. My observation recommendations are as follows;
- Author (s) should improve the contribution section. They need to explain what is the specific contribution of this study to the overall hypothesis. – thank you for the invaluable comment. The contribution section was improved (Line 84-91).
-line 292- PM is not defined, AP is not given in any equation. – sorry that was a typographical error. AP was replaced with PM.
Table 1- add description of each variable used in the model – we correct this; description of the variables was done in Table 1.
line 312- cite studies to support these criteria - the basis for the selection of variables and the model was indicated in lines 240-241 and 282-283. On top of this, we also added a paragraph (line 303-318) that explained how the variables were selected.
- I am surprised to know that the Level of Particulate Matter (PM2.5) is not significantly related to life expectancy, however, the author (s) has provided a possible reason for this by mentioning good health expectancy. This must be discussed in the limitation of the study. – yes we had raised this in the conclusion section.
-How infant mortality was operationalised should be discussed, because it is creating confusion while looking at the results of the impact of health expenditures on Life expectancy and infant mortality, it should have the opposite effect on both. The results show a positive effect on both. – the effect of health expenditures on life expectancy is insignificant. Which means that health expenditure is not influencing Life expectancy. We explained this and cited studies which showed that much of the health expenditure does not really reach the intended targets. However, health expenditure is seen to be improving infant mortality. This is true in developing countries because governments is these countries focus more on primary health care. It may be that they are channelling a lot of funds in primary healthcare.
-line 338- how results are in line with the studies cited here, almost all studies cited do not mention child death as a consequence of PM emission. while this study shows a positive association between infant mortality and PM level. - we managed to add more studies that explained the effects of air pollution on children (line 384-394).
-limitation, future directions, and implication section must be added in revised version. – thank you. We had included these in the conclusion section.